# Beverage Consumption Patterns and Nutrient Intake Are Associated with Cardiovascular Risk Factors among Urban Mexican Young Adults

**DOI:** 10.3390/nu15081817

**Published:** 2023-04-09

**Authors:** Rocío Guadalupe Salinas-Mandujano, Estefany Laiseca-Jácome, Minerva Ramos-Gómez, Rosalía Reynoso-Camacho, Luis Miguel Salgado, Miriam Aracely Anaya-Loyola

**Affiliations:** 1Research and Graduate Studies in Food Science, School of Chemistry, Autonomous University of Queretaro, Queretaro 76010, Mexico; 2Graduate Studies in Human Nutrition, Department of Natural Science, Autonomous University of Queretaro, Queretaro 76230, Mexico; 3Centro de Investigación en Ciencia Aplicada y Tecnología Avanzada, Instituto Politécnico Nacional, Querétaro 76090, Mexico

**Keywords:** beverage consumption patterns, nutritional intake, cardiovascular risk factors, Mexican young adults

## Abstract

Regular consumption of sugar-sweetened beverages has been related to metabolic diseases. Our objective was to identify beverage consumption patterns, nutrient intake, and their possible association with the prevalence of cardiovascular risk factors among Mexican young adults. A cross-sectional survey was conducted. Beverage consumption patterns were obtained by principal components analysis. Logistic regression models were applied to assess the association between the beverage patterns and cardiovascular risk factors. Four beverage patterns were identified. Higher consumption of alcoholic beverages was associated with lower odds for high body fat percent (OR: 0.371; 95% CI: 0.173–0.798), high blood pressure (OR: 0.318; 95% CI: 0.116–0.871), and high glucose (OR: 0.232; 95% CI: 0.061–0.875). Higher consumption of yogurt was associated with lower odds for high glucose (OR: 0.110; 95% CI: 0.22–0.559). In contrast, highest consumption of juice had greater odds for high triglycerides (OR: 1.084; 95% CI: 1.011–4.656). Higher consumption of milk was associated with greater odds for high glucose (OR: 5.304; 95% CI: 1.292–21.773). Beverage consumption habits in Mexican young adults are associated with increased risk factors for cardiovascular disease. Therefore, intervening during young adulthood should be considered in order to improve current health and prevent cardiovascular mortality in later decades.

## 1. Introduction

Cardiovascular diseases (CVD) are the most common causes of death worldwide, and over 80% of those deaths take place in low- and middle-income countries [1]. Specifically in Mexico, The Global Burden of Disease study estimated that CVD accounted for 22.7% of all deaths. From 2007 to 2017, deaths from CVD increased 51.1%, where trends in mortality had a significant increase in young adults (<35 years old) [1,2]. 

The increasing rates of CVD in Mexico have been attributed to the high prevalence of cardiovascular risk factors (CVRFs), such as obesity (75.2%) [3], high blood pressure (BP) (49.4%) [4], high density lipoprotein cholesterol (HDL-C) (52.3%), high triglycerides (TG) (32.3%), total cholesterol (TC) (19.2%), low density lipoproteins cholesterol (LDL-C) (41.7%), physical inactivity (17.4%), and tobacco smoking (19.9%) [5,6]. Among them, National health surveys in Mexico have identified a body mass index (BMI) above 25 kg/m^2^, and dyslipidemias as the most prevalent modifiable CVRFs in adults [7]. The prevalence of obesity, Type 2 Diabetes Mellitus (T2DM), and stagnation in the improvement of lipid levels in young adults, all are in the setting of suboptimal lifestyle behaviors, such as decreased physical activity and increased rates of smoking, as well as the adoption of poor diet characterized by a high caloric intake [8,9].

Sugar-sweetened beverages (SSBs) are among the dietary components that contribute most to the total energy intake in the Mexican population [10]. Mexican adults (>20 years old) have reported an estimated daily consumption of 415 kcal from all beverages, representing a contribution of 22.5% to the energy intake. Alcohol, whole and flavored milk, and soda have been the beverages that have contributed most to the energy intake (16.17, 8.25, 8.04 and 6.60%, respectively) [11]. SSBs are considered the main source of sugars in the Mexican diet, and have been associated with a low intake of important micronutrients, such as vitamin C, vitamin A, riboflavin, magnesium, calcium, and fiber [12,13].

Studies have examined the beneficial or deleterious effects of specific beverage consumption on weight gain, T2DM, and CVRFs among different young populations. A prospective, randomized, controlled crossover trial demonstrated that interventions with low to moderate SSBs significantly increased BMI, waist circumference (WC), GLC, and significantly reduced LDL particle size. In addition, carbohydrate intake increased, while protein, fat, and β-Carotene intake decreased in SSB interventions among Swiss young men [14]. A prospective study reported that higher SSB consumption was associated with higher risk of high WC, high LDL-C, high TG, and hypertension. Whole-fat milk consumption was associated with lower risk of high TG. Consumers of both whole-fat milk and SSBs were more likely to have higher total energy intake, to be current smokers, and to have lower levels of physical activity [15]. A cross-sectional study showed that a greater dairy consumption was associated with a higher prevalence of obesity, a lower prevalence of high BP, greater calorie intake, and greater intake of carbohydrates, protein, total fats, cholesterol, fibers, sodium, calcium, and phosphorous in Brazilian young adults [16]. Du el al. reported that light consumption of beer was associated with lower WC and higher HDL-C levels, and high consumption of wine with higher BP in Australian young adults [17]. Hillesund et al., reported a positive association between increasing alcohol intake and prevalence of obesity, calorie intake, and nutrient intake in both young women and men. An inverse association with energy-adjusted intake of thiamine, phosphate, Fe, Zn and Se in young men, and with vitamin A, β-carotene, vitamin E and C, thiamine, vitamin B6, folate, P, Mg, K, Fe, Zn, and Cu in young women [18]. 

Most studies focused only on a specific beverage item rather than on the comprehensive consumption pattern of beverages in order to evaluate the risk of obesity and CVRFs. Given that Mexico is the country with the largest consumption of soft drinks, it is possible that these might represent an important contribution to the increase in the obesity and CVD epidemics currently taking place [11,19]. However, a large number of beverages have been introduced into diet, which are either sugar-sweetened or caffeinated, and often both, but many have emerged with reduced caloric levels, a range of flavors, and often caffeine and other food constituents have been added [20]. Therefore, additional studies should examine the beverage consumption patterns rather than consumption of individual beverages in relation to total energy intake, obesity, and CVRFs. 

In our study, the hypothesis tested was that beverage consumption patterns with positive high factor loading scores for SSBs would be associated with greater energy intake, obesity, and CVRFs when compared to those with low factor loading scores of SSBs. Therefore, the aims of this study were to identify beverage consumption patterns and to examine their association with energy and nutrient intake, as well as to evalaute the prevalence of obesity and CVRF’s among Mexican young adults from urban areas. 

## 2. Materials and Methods

### 2.1. Study Population

The data for this study were drawn from a cross-sectional epidemiological survey conducted among students from the Autonomous University of Queretaro (UAQ), Mexico between January and November of 2014. Subjects (n = 1160) in the study were recruited by a non-probabilistic sample, including all the freshmen attending the Autonomous University of Queretaro, who voluntarily agreed to participate in the study. All participants gave their written informed consent according to the Helsinki Declaration. The study was approved by the Ethics Committee at the School of Natural Sciences of the University of Queretaro (23FCN, 2014).

The inclusion criteria used were students in their first year of school between 18 and 25 years of age. The exclusion criteria were participants with current clinical evidence of infectious disease, anemia, pregnancy or lactation, current medical or nutritional treatment, as well as those with missing information derived from a dietary or beverage questionnaire, biochemical measurements, and anthropometric measurements. The final sample consisted of 340 young Mexican adults (184 women and 156 men).

All participants completed detailed food frequency and beverage frequency questionnaires and underwent an extensive clinical and nutritional evaluation including physical examination, fasting blood chemistry analyses, and anthropometric and body composition.

### 2.2. Dietary and Beverage Consumption Assessment

#### 2.2.1. Dietary Assessment

A semi-quantitative food frequency questionnaire (FFQ) was used to assess diet through collecting data on the frequency of consumption of 160 food items. For each food item, a commonly used portion size (e.g., one slice of bread) was specified, and participants were asked how frequently they had consumed the food over the previous year. They were able to choose from ten responses ranging from ‘never’ to ‘six or more times per day’. The total energy, the total consumption of protein, fat, and carbohydrates, as well as their percentage of total energy, was estimated by multiplying the frequency of consumption of each food type included in the FFQ by the nutrient content estimated by a comprehensive database of food contents. The adequacy of macronutrient intake was assessed according to the Institute of Medicine (IOM) committee recommendations [21].

#### 2.2.2. Beverage Assessment

We obtained beverage intake data by using a beverage frequency questionnaire (BFQ) of 50 items. Participants were asked how often they had consumed each beverage over the previous year. They choose from five responses ranges: “never”, “per year”, “per month”, “per week”, and “per day”. Participants were able to choose the beverage portion size from a pre-set list with pictures of various sizes of mugs, glasses, and bowls. Then, we converted portion sizes into volume (mL). 

#### 2.2.3. Beverage Patterns

Cronbach’s alpha coefficient was used to assess the reliability of BFQ scores. The Cronbach´s alpha value was 0.819, which was considered good internal consistency (α ≥ 0.7) [11]. Items identified as unnecessary were removed from the scores (n = 22 items).

To estimate energy and nutrient intake by beverage, parameters were calculated using food composition data from the United States Department of Agriculture (USDA) [22], and polyphenol content was calculated using the Phenol-Explorer 3.0 [23]. 

Since most of the amounts of consumed beverage distributions were skewed toward higher values, all variables were log- (natural) transformed and then energy-adjusted using the residual method [24], which uses the residual from regression models with beverage intake as the dependent variable and energy intake as the independent variable. 

Principal components analysis (PCA) with varimax rotation was performed on the beverage-items transformed and adjusted beverages. The patterns described by each component are direct linear relationships between the underlying beverage variables and they may be interpreted by their factor loadings [25]. Large positive or negative factor loadings indicates the beverage that is important in that component. Factors that have eigenvalues greater than 1 were retained for interpretation. Beverages with factor loading values ≥ 0.3 were defined as strong association and used to characterize each pattern. The scores were categorized into tertiles and labelled according to the beverage with high loadings. 

### 2.3. Anthropometric Measurements and Blood Pressure

A trained nutritionist measured all anthropometric variables. Height was measured to the nearest 0.1 cm using a mechanic stadiometer (SECA Mod 216, Hamburg, Germany). Body weight and body fat percent (BF %) were measured simultaneously by using a digital scale Body Composition Analyzed X-Scan plus II (Jawon Medical Mod 514 Co., Ltd., Seoul, Republic of Korea). WC was measured midway between the lowest rib and the iliac crest using flexible fiberglass measuring bands (SECA Mod 200, Hamburg, Germany) and Body mass index (BMI) was calculated from weight and heigh values according to Quetlet´s score. The BP was recorded 3 times with a 10 min interval between measurements by using a mercurial sphygmomanometer (Medimetrics Mod 5881).

### 2.4. Biochemical Measurements

Venous blood samples were collected after 8–12 h of overnight fasting. Standard enzymatic colorimetric methods (Spinreact, Girona, Cataluña, Spain) were used to assess serum GLC, TC, TG, LDL-c, and HDL-c concentrations in a Mindray BS120 biochemical automatized analyzer (Shenzhen, China). 

### 2.5. Definition of Cardiovascular Risk Factors

High BMI was defined as ≥25. High BF% was defined according to Gallagher [26] as high BF% (≥20% in men, and 31, 32, and 33% in 18, 19, and ≥20-year-old women, respectively). According to the International Diabetes Federation [27], High BP was defined as ≥130 mmHg of systolic blood pressure (SBP), and ≥85 mmHg of diastolic (DBP); high GLC ≥ 100 mg/dL, high TC ≥ 200 mg/dL, high TG ≥ 100 mg/ dL, high LDL-C ≥ 100 mg/dL, and low HDL-C ≤ 40 and 50 mg/dL, in men and women, respectively.

### 2.6. Covariates

Important confounders of the association between beverage consumption and CVRF´s were considered. Among them, smoking status was classified as two groups: nonsmoker or current smoker. Physical activity was estimated by mean of a 24 hr recall questionnaire [28]. According to the Recommendations for adults from the American College of Sport Medicine (ACSM) and the American Heart Association (AHA) [29], they were categorized into two groups: a sedentary group when young people practiced less than 150 min of physical activity in a week, and a physically active group. Gender was classified into two groups: women and men. 

### 2.7. Data Analyses

All analyses were performed with α = 0.05 significance level using SPSS version 25.0 (SPSS Inc., Chicago, IL, USA). Continuous variables were reported as mean ± standard error (SE). Categorical variables were expressed as frequencies with weighted percentages. ANOVA were conducted to test for significant differences in mean values of anthropometric, biochemical, and nutritional variables, with Bonferroni-adjustment for multiple comparisons. The Chi-square (χ^2^) test was performed in order to compare the prevalence of covariates across tertiles of beverage consumption patterns scores. To assess the association between beverage consumption patterns and the risk of CVRFs, multivariate logistic regression models were applied. Odds ratios (OR) and 95% confidence intervals (CI) were calculated using the following models: Model 1 was not adjusted for any covariate; model 2 was adjusted for gender, smoking status, physical activity, and BMI; model 3 adjusted all covariates in model 2 plus total energy intake and percentage of energy from protein, fat, and carbohydrates.

## 3. Results

### 3.1. Beverage Patterns Description

PCA identified 4 major factors according to the eigenvalue >1.0 criterion applied. These factors accounted for 33.05% of the variance in the beverage (Table 1). Pattern 1, which explained 18.44% of the variance in the sample and was labeled “Alcoholic drink pattern”, showed positive high factor loading scores for tequila, rum, brandy, vodka, whisky, beer, red and white wine, and carbonated and tonic water. Pattern 2 explained 9.38% of the variance and was labeled “Yogurt drink pattern” due to the high positive factor loading on drinkable plain yogurt, drinkable flavored yogurt, and lactic fermented dairy drink. Pattern 3 explained 6.90% of the variance and was labeled “Juicy pattern” because of the positive high factor loading scores with bottled or canned fruit juice, bottled or canned fruit nectar juice, natural juice, and soy juice. Finally, based on the positive high factor loading scores on milkshakes with fruit, milkshake or Skimoo with milk, and atole with milk, pattern 4 explained 6.145 of the variance and was labeled “Milk beverage pattern”.

### 3.2. Association of Beverage Consumption Patterns and Energy and Nutrient Intake

Mean total energy and nutrient intake according to tertiles of the beverage consumption pattern scores are presented in Table 2. Total protein, total fat, and total carbohydrates are presented as percentages of energy intake, whereas energy from beverages, protein, fat, carbohydrates, fiber, vitamins, minerals, and polyphenols are given as nutrient intake. Increasing scores of the Alcoholic drink pattern were positively associated with higher protein (20.09 ± 1.93), fat, saturated fat (11.43 ± 1.19), and cholesterol (59.14 ± 6.23) intake, as well as calcium (655.97 ± 62.4), vitamin A (242.98 ± 24.5), vitamin D (4.59 ± 0.42), anthocyanins (7.88 ± 1.28), phenolic acids (2.58 ± 0.28), stilbenes (2.58 ± 0.28), and other polyphenols (7.17 ± 0.75). 

Total energy from beverages (788.38 ± 46.61), saturated fat (9.32 ± 0.82), and cholesterol (49.17 ± 4.33) intake increased significantly from tertile 1 to tertile 2 of the Yogurt pattern. Thiamine (0.3 ± 0.03), caffeine (52.56 ± 3.93), anthocyanins (7.26 ± 1.29), and flavonoids (96.43 ± 10.02) intake was significant higher with increasing Yogurt pattern scores. 

With increasing scores of the Juicy pattern, energy from beverages (796.81 ± 48.1), caffeine (64.47 ± 5.26), anthocyanins (7.35 ± 1.27), flavonoids (88.01 ± 9.17), phenolic acids (1.38 ± 0.25), and stilbenes (1.38 ± 0.25) intake significantly increased with increasing scores of the Juicy pattern. 

As for the Juicy pattern, energy from beverages (788.61 ± 47.1), caffeine (59.97 ± 4.36), anthocyanins (6.91 ± 1.28), flavonoids (85.15 ± 9.08), phenolic acids (1.53 ± 0.28), and stilbenes (1.53 ± 0.28) intake increased with significant increasing scores of the Milk beverages pattern.

### 3.3. Difference of Anthropometric and Biochemical Parameters among Beverage Patterns

Characteristics of Mexican young adults according to beverage consumption pattern scores are presented in Table 3. For the Alcoholic pattern, SBP (113.28 ± 1.15) and TG levels (52.84 ± 1.43) increased significantly in the highest tertile. Compared with participants in the lowest tertile, participants in the highest tertile of Yogurt Pattern had higher TG (104.33 ± 6.06) and TC levels (167.24 ± 2.94). For both Juicy pattern and Milk beverage pattern, all anthropometric and biochemical parameters remained constant across tertiles.

### 3.4. Difference of Covariables among Beverage Patterns

Covariates of Mexican young adults are shown in Table 3. In general, the Alcoholic drink pattern was the only pattern that showed a statistical difference of covariates across tertiles. Compared with individuals in the lower tertile, those in the highest tertile were more likely to be men (59.1%) and to smoke (60.9%).

### 3.5. Association of Beverage Patterns and Cardiovascular Risk Factor

Unadjusted and adjusted ORs for the cardiovascular risk factor among tertiles of beverage patterns are presented in Table 4. In the unadjusted model, compared to the first tertile, the highest tertile of the Alcoholic pattern was associated with reduced odds of BF% (OR: 0.570; 95% CI: 0.335–0.971). When age, smoking status, physical activity, and BMI were controlled, this association disappeared, but then, participants in the highest tertile showed reduced odds of high BF% (OR: 0.371; 95% CI: 0.173–0.798) than those in the lowest tertile. Subsequently, when total energy intake and percentage of energy from protein, fat, and carbohydrates were also controlled, individuals in tertile 2 showed more reduced odds of high GLC (OR: 0.232; 95% CI: 0.061–0.875) than those in the lowest tertile. However, the association of this beverage pattern with high BF% disappeared.

After adjusting for gender, smoking status, and BMI, lower odds of high GLC (OR: 0.317; 95% CI: 0.102–0.987) were observed in tertile 2 of the Yogurt pattern. After additionally adjusting for total energy intake and percentage of energy from protein, fat, and carbohydrates, this association was attenuated but remained statistically significant (OR: 0.110; 95% CI: 0.022–0.559). 

We also founded that controlling for age, smoking status, physical activity, and BMI, subjects in tertile 2 of the Juicy pattern had higher odds of high TG (OR: 1.155; 95% CI: 1.0.11–1.656) than those in the lowest tertile. Even when additionally controlling for total energy intake and percentage of energy from protein, fat, and carbohydrates, this association was attenuated but remained statistically significant (OR: 1.084; 95% CI: 1.011–1.656). 

Finally, tertile 2 of the Milk beverage pattern was associated with increased odds of high GLC (OR: 5.304; 95% CI: 1.292–21.773) when age, smoking status, physical activity, and BMI, total energy intake and percentage of energy from protein, fat, and carbohydrates were controlled.

## 4. Discussion

Multiple nutritional epidemiology studies suggest that lifestyle, focused on diet, significantly influences CVD occurrence [30], and it may be particularly important in young adults, because the prevalence of several CVRFs has increased among subsets of young adults [31,32].

In this study, four major beverage consumption patterns were identified: Alcoholic drink pattern, Yogurt drink pattern, Juicy drink pattern, and Milk beverages pattern, and differences among them took into consideration both beverage preferences and their nutrimental composition. The Milk and Juicy patterns showed a positive association with CVRFs, whereas the Alcohol and Yogurt patterns showed an inverse association after adjusting for age, smoking status, physical activity, BMI, total energy intake, and percentage of energy from protein, fat, and carbohydrates. Our findings are in line with Rivera et al. [33]. They also identified four major beverage consumption patterns in Mexican adults labeled as Alcohol, Coffee/tea, Soft drinks and Low-fat milk. The first pattern was similar with that obtained in the current study, whereas the last three were different. It was expected that beverage consumption patterns would not be completely in accordance with each other since they were extracted from data obtained in studied populations with different age ranges and morbidities. 

Despite the high intake of different types of SSBs, such as fruit drinks, sweetened milk and milk alternatives [13], mainly characterized the Milk, Yogurt, and Juicy patterns, respectively, only the Milk pattern was associated with high risk of GLC. Dietary intervention studies investigating the effects of milk or dairy products and glucose response indicate a wide range of effects, which are likely due to the different types of dairy utilized [34]. Concerning milk and fermented dairy beverages, previous studies have suggested favorable effects of dairy intake on high GLC and the risk of T2DM. For example, epidemiological studies suggest that chronically high consumption of milk is associated with a reduced risk of T2DM. These effects can be hypothetically explained by increased insulinemic response, decreased glycemic fluctuations, and increased secretion of GIP and GLP-1 triggered by milk proteins [35]. Likewise, in the Prevention with Mediterranean Diet (PREDIMED) study, a prospective study conducted on community-dwelling elderly nondiabetics, subjects at high cardiovascular risk observed an inverse association between both low-fat and whole-fat yogurt and the risk of T2DM. However, yogurt consumption was associated with a reduction in risk only when one serving of a combination of biscuits and chocolate confectionary or whole grain biscuits and homemade pastries was replaced with one serving of yogurt. A possible explanation is that, although nutritionally yogurt is comparable to milk, processing, added ingredients, and fermentation improve the nutritional value of yogurt and provide it with unique properties that enhance the bioavailability of flavonoids [36]. Moreover, data from the Maastricht Study showed significant inverse associations of fermented dairy products and yogurt with high GLC [36]. Another cross-sectional study performed among Dutch adults showed inverse associations of fermented dairy products with pre-diabetes [37].

These studies are consistent with our findings that showed a significant inverse association between the Yogurt pattern and high risk of GLC, which can be attributed to the association of this beverage pattern with the high intake of anthocyanins and flavonoids. In contrast, in our current study, we observed that the Milk pattern showed a positive association with high GLC, which was expected because of the association of this beverage pattern with high energy consumption due to SSBs, such as milkshake, Skimoo, and atole, which mainly characterized this beverage pattern, instead of whole-fat and low-fat isolated.

Consistently, previous studies have observed similar results regarding SSBs and a high risk of GLC among different populations. For instance, findings from prospective cohort studies support a strong positive link between the intake of SSBs and the risk of T2DM, in part by incomplete compensation for energy at subsequent meals following intake of liquid calories, and also by contribution to a high dietary glycemic load leading to inflammation, insulin resistance, and impaired β-cell function [38]. Similarly, a review paper, that included searches for clinical and observational studies, provided evidence to prove a positive association between regular SSBs and weight gain, and also the eventual risk for T2DM, due to SSBs, that can alter glucose handling and insulin sensitivity [39]. In another prospective cohort study in the U.S. among female nurses aged 24 to 44 years, high SSB consumption was positively associated with the risk of T2DM [40]. Data from a prospective cohort study in middle-aged adults suggested that regular SSB intake is associated with increased insulin resistance and a greater risk of developing prediabetes [41]. Additionally, the Health Workers Cohort Study in the Mexican states of Morelos and Mexico reported that subjects consuming >2 servings of sweetened beverages daily are at 2.1 times greater risk of raised fasting plasma GLC compared with non-consumers [42]. Likewise, a cross-sectional study among young rural adults in South Africa found that the consumption of SSBs is significantly associated with increased high risk of high GLC [43]. 

Concerning the Juicy pattern, fruit juices are still widely perceived as a healthier option than SSBs, primarily because they are related to high antioxidant and bioactive substances (including vitamins, minerals, and polyphenols) which have been reported to be protective against adverse health effects [44]. However, the primary components of fruit juices, after water, are free sugars. Among them, the fructose content of most natural fruit juice is quite similar to that of beverages sweetened with high-fructose corn syrup, such as commercial fruit juice, which can induce hypertriglyceridemia [45,46].

In our study, the Juicy pattern was inversely associated with high TG, whereas it was positively associated with high energy intake. In agreement, Stanhope et al. provides evidence that postprandial TG was significantly increased in response to a 2 week consumption of 25% of energy as high fructose corn syrup in young, normal-weight, and overweight subjects [47]. The mechanism proposed for this association is that the excessive intake of fructose contained in commercial fruit juices might lead to a significantly enhanced rate of de novo lipogenesis and triglyceride synthesis [48].

In our present study, the Alcoholic pattern showed an inverse association with risk of high BP and high GLC, and a positive association with intake of anthocyanins, phenolic acids, stilbenes and other polyphenols. This is consistent with epidemiological evidence that has pointed out that moderate alcohol consumption is inversely associated with CVRFs [49]. A Japanese cohort of people aged 40–55 years found that light to moderate alcohol consumption was associated with a lower risk of T2DM compared to non-drinkers [50]. Likewise, a review report indicated that daily moderate consumption of alcohol beverages was inversely related to the development of hypertension [51]. The underlying mechanism to explain these protective effects has been attributed to the ethanol content present in all alcoholic beverages, as well as the polyphenolic content present in beer and red wine [52]. Studies have found that polyphenols in wine and beer decrease blood pressure while increasing plasma nitric oxide concentration [53]. The cardio protective effect of moderate alcohol consumption in relation to CVRFs involves a beneficial influence on reverse cholesterol transport, systemic inflammation and oxidative stress, endothelial function and platelet aggregation, reduction in body fat, improvement of insulin sensitivity, and modulation of gene expression involved in inflammation and cholesterol synthesis [54].

The results of the present study might indicate that a Mexican young adults’ preference of beverages is based on taste instead of health considerations since they prefer alcoholic and SSBs, which are related to poor in nutrient intake and high energy density.

Therefore, it is imperative to carry out interventions that provide the health effects of nonalcoholic beverages, such as caloric soft drinks, noncaloric soft drinks, coffee and tea, energy drinks, sport drinks, kombucha, and sparkling water, in order to address the existing nutrient gap, increase phytonutrient intake, and reduce risk of chronic disease [55].

To our knowledge, this is the first cross-sectional epidemiological study with the goal of identifying beverage consumption patterns among urban Mexican young adults. In addition, it provides useful information to elucidate the effect of beverage intake pattern on CVRFs among Mexican young adults. The results suggest performing similar studies in different populations because beverage preferences are dependent on cultural and socio-economics factors, such as smoking status and physical activity. One limitation of the current study was that since it was a cross-sectional design study, the current results cannot provide insight on the causation between beverages patterns and CVRF´s. Subsequent prospective studies are needed to support the findings of the current study. 

## 5. Conclusions

Beverage consumption habits in Mexican young adults are associated with increased risk factors for cardiovascular disease. Therefore, intervening during young adulthood should be considered in order to improve current health and prevent cardiovascular mortality in later decades. However, further longitudinal studies are required to confirm the current results. 

## Figures and Tables

**Table 1 nutrients-15-01817-t001:** Assessment of the factor-loading matrix for the four dietary patterns among Mexican young.

	Alcoholic Drink Pattern	Yogurt Drink Pattern	Juicy Pattern	Milk Beverages Pattern
Tequila, rum, brandy, vodka, whisky	0.897			
Beer	0.880			
Drinkable plain yogurt		0.907		
Drinkable flavored yogurt		0.835		
Lactic fermented dairy drink		0.566		
Bottled or canned fruit juice			0.853	
Bottled or canned fruit nectar juice			0.798	
Natural juice			0.549	
Milkshake with fruit				0.665
Milkshake or Skimoo with milk				0.557
Carbonated water	0.458			0.535
Atole with milk				0.527
Black coffee				
Tea				
Powdered flavored water				
Boted flavored water				
Bottled coffee				
Soy juice			0.310	
Tonic water	0.338			
Flavored milk				
Sport drink				
Red and White wine	0.469			
Bottled tea				
Smoothie				
Percentage of variability explained	18.44	9.38	6.90	6.14

Factor loadings are equivalent to simple correlations between food items and dietary patterns, and those less than 0.30 were not shown for simplicity.

**Table 2 nutrients-15-01817-t002:** Mean total energy and nutrient intake according to tertiles of the beverage consumption pattern scores identified among Mexican young adults.

	Alcoholic Drink Pattern			
	Tertile 1	Tertile 2	Tertile 3	*p*-Value ^a^
**Total dietary intake**				
Total energy requirement	2025.50 ± 41.54	1987.44 ± 37.39	2026.57 ± 37.04	0.717
Total energy intake (Kcal)	2798.17 ± 130.90	2935.83 ± 165.04	3183.70 ± 154.43	0.194
Total protein (%)	17.75 ± 0.71	16.87 ± 0.60	17.27 ± 0.55	0.600
Total fat (%)	29.70 ± 0.88	30.16 ± 0.87	29.25 ± 0.93	0.891
Total carbohydrates (%)	51.26 ± 1.30	52.09 ± 1.20	51.53 ± 1.23	0.779
Total fiber (g)	46.90 ± 5.42	42.31 ± 4.32	43.74 ± 3.25	0.757
Sugar (g)	21.98 ± 1.05	22.66 ± 1.06	22.28 ± 1.05	0.895
**Energy and nutrient intake from beverages**				
Total energy (Kcal)	570.57 ± 46.31	580.03 ± 34.98	605.43 ± 31.16	0.799
Energy (%)	23.31 ± 2.36	20.15 ± 1.40	22.42 ± 1.79	0.470
Protein (g)	13.9 ± 1.29	16.51 ± 1.49	20.09 ± 1.93	**0.024**
Carbohydrate (g)	102.01 ± 7.91	120.46 ± 9.15	113.2 ± 9.93	0.348
Fiber (g)	1.18 ± 0.13	1.45 ± 0.19	1.31 ± 0.21	0.560
Sugar (g)	98.94 ± 7.76	113.96 ± 8.57	110.51 ± 9.62	0.441
Fat (g)	11.66 ± 1.12	15.03 ± 1.5	18.68 ± 1.92	**0.006**
Saturated fat (g)	7.07 ± 0.71	9.14 ± 0.93	11.43 ± 1.19	**0.007**
Monounsaturated fat (g)	1.52 ± 0.26	1.3 ± 0.16	1 ± 0.12	0.159
Polyunsaturated fat (g)	0.23 ± 0.03	0.22 ± 0.02	0.18 ± 0.02	0.267
Cholesterol (mg)	38.36 ± 3.74	49.23 ± 4.93	59.14 ± 6.23	**0.016**
Na (mg)	271.73 ± 21.51	342.56 ± 25.04	344.89 ± 28.77	0.069
K (mg)	657.71 ± 64.46	629.14 ± 54.49	586.76 ± 66.78	0.720
Ca (mg)	436.38 ± 40.51	525.55 ± 47.62	655.97 ± 62.4	**0.010**
Mg (mg)	45.6 ± 4.54	43.25 ± 3.75	39.76 ± 4.45	0.623
Fe (mg)	42.87 ± 2.45	0.81 ± 0.08	1.14 ± 0.14	0.158
Zn (mg)	25.03 ± 8.81	25.03 ± 8.81	13.93 ± 4.98	0.183
P (mg)	229.34 ± 31.73	206.23 ± 21.42	169.77 ± 19.46	0.232
Se (μg)	6.11 ± 0.8	5.69 ± 0.58	4.7 ± 0.52	0.285
Vit C (mg)	66.96 ± 6.63	73.19 ± 7.98	69.7 ± 8.18	0.846
Thiamin (mg)	0.28 ± 0.03	0.43 ± 0.1	0.34 ± 0.06	0.311
Riboflavin (mg)	0.49 ± 0.05	0.41 ± 0.04	0.39 ± 0.05	0.267
Niacin (mg)	3.3 ± 0.37	4.2 ± 0.37	3.08 ± 0.29	0.055
Vit B6 (mg)	0.35 ± 0.04	0.29 ± 0.03	0.32 ± 0.04	0.524
Folate (μg)	39.57 ± 4.05	42.92 ± 4.56	40.8 ± 4.48	0.860
Choline (mg)	43.52 ± 4.69	40.58 ± 3.54	36.13 ± 3.92	0.437
Vit B12 (mg)	0.99 ± 0.13	0.83 ± 0.09	0.67 ± 0.08	0.073
Vit E (mg)	0.27 ± 0.03	0.28 ± 0.03	0.28 ± 0.04	0.954
Vit A (μg)	150.98 ± 12.05	195.77 ± 18.86	242.98 ± 24.5	**0.003**
Vit D (mg)	2.44 ± 0.19	3.04 ± 0.29	4.59 ± 0.42	**<0.0001**
Caffeine (mg)	50.13 ± 5.33	44.89 ± 3.18	51.91 ± 3.45	0.452
Anthocyanins (mg)	2.68 ± 0.33	3.32 ± 0.48	7.88 ± 1.28	**<0.0001**
Flavonoids (mg)	59.7 ± 5.3	67.67 ± 7.27	79.78 ± 7.93	0.121
Phenolic acids (mg)	0.1 ± 0.02	0.38 ± 0.05	2.58 ± 0.28	**<0.0001**
Stilbenes (mg)	0.1 ± 0.02	0.38 ± 0.05	2.58 ± 0.28	**<0.0001**
Other polyphenols (mg)	3.8 ± 0.3	4.8 ± 0.39	7.17 ± 0.75	**<0.0001**
	**Yogurt pattern**			
**Total dietary intake**				
Total energy requirement	2001.07 ± 39.89	1963.81 ± 35.36	2074.84 ± 40.11	0.117
Total energy intake (Kcal)	3084.72 ± 155.05	3002.16 ± 157.07	2809.58 ± 140.35	0.407
Total protein (%)	17.88 ± 0.54	17.14 ± 0.63	16.88 ± 0.72	0.507
Total fat (%)	30.49 ± 0.83	29.12 ± 0.85	29.53 ± 1.01	0.532
Total carbohydrates (%)	51.46 ± 1.28	52.37 ± 1.08	51.45 ± 1.39	0.733
Total fiber (g)	38.36 ± 3.89	46.66 ± 4.69	48.38 ± 4.96	0.246
Sugar (g)	21.97 ± 1.04	22.45 ± 1.05	22.49 ± 1.08	0.928
**Energy and nutrient intake from beverages**				
Total energy (Kcal)	403.68 ± 25.03	564.12 ± 29.77	788.38 ± 46.61	**<0.0001**
Energy (%)	17.34 ± 1.2	19.98 ± 1.46	29.2 ± 2.67	**<0.0001**
Protein (g)	14.13 ± 1.25	19.33 ± 2.02	17.01 ± 1.42	0.071
Carbohydrate (g)	96.64 ± 7.27	123.42 ± 10.65	115.59 ± 8.74	0.098
Fiber (g)	1.07 ± 0.15	1.67 ± 0.23	1.2 ± 0.14	0.043
Sugar (g)	92.25 ± 6.84	119.81 ± 10.37	111.3 ± 8.3	0.071
Fat (g)	12.36 ± 1.19	17.72 ± 2.02	15.27 ± 1.31	0.052
Saturated fat (g)	7.47 ± 0.74	10.84 ± 1.26	9.32 ± 0.82	**0.050**
Monounsaturated fat (g)	1.11 ± 0.16	1.4 ± 0.19	1.31 ± 0.22	0.557
Polyunsaturated fat (g)	0.18 ± 0.02	0.23 ± 0.03	0.21 ± 0.03	0.416
Cholesterol (mg)	39.81 ± 3.81	57.68 ± 6.64	49.17 ± 4.33	**0.047**
Na (mg)	277.23 ± 20.75	353.04 ± 31.63	328.82 ± 22.01	0.098
K (mg)	559.31 ± 47.85	717.25 ± 70.59	596.28 ± 64.62	0.169
Ca (mg)	454.92 ± 40.62	616.11 ± 64.5	546.06 ± 45.37	0.085
Mg (mg)	38.37 ± 3.35	48.22 ± 4.68	41.97 ± 4.57	0.253
Fe (mg)	0.81 ± 0.1	1.22 ± 0.16	0.98 ± 0.11	0.080
Zn (mg)	9.94 ± 2.42	13.55 ± 3.28	25.31 ± 9.59	0.169
P (mg)	178.11 ± 20.25	224.28 ± 25.33	202.8 ± 28.22	0.421
Se (μg)	4.9 ± 0.55	6.02 ± 0.66	5.57 ± 0.72	0.469
Vit C (mg)	65.88 ± 6.95	78.91 ± 8.91	65.02 ± 6.76	0.351
Thiamin (mg)	0.26 ± 0.02	0.49 ± 0.11	0.3 ± 0.03	**0.044**
Riboflavin (mg)	0.36 ± 0.03	0.49 ± 0.05	0.44 ± 0.05	0.121
Niacin (mg)	3.4 ± 0.4	3.57 ± 0.34	3.61 ± 0.31	0.901
Vit B6 (mg)	0.25 ± 0.02	0.38 ± 0.05	0.33 ± 0.04	0.060
Folate (μg)	38.46 ± 4.03	47 ± 5.1	37.8 ± 3.81	0.250
Choline (mg)	35.5 ± 3.22	44.98 ± 4.37	39.7 ± 4.49	0.257
Vit B12 (mg)	0.71 ± 0.08	0.9 ± 0.1	0.89 ± 0.12	0.339
Vit E (mg)	0.25 ± 0.03	0.29 ± 0.03	0.29 ± 0.04	0.669
Vit A (μg)	165.03 ± 15.55	225.76 ± 24.59	198.67 ± 16.46	0.085
Vit D (mg)	2.92 ± 0.29	3.68 ± 0.4	3.46 ± 0.27	0.230
Caffeine (mg)	37.68 ± 3.23	56.58 ± 4.77	52.56 ± 3.93	**0.003**
Anthocyanins (mg)	3.28 ± 0.41	3.34 ± 0.42	7.26 ± 1.29	**0.001**
Flavonoids (mg)	55.93 ± 4.42	54.89 ± 4.03	96.43 ± 10.02	**<0.0001**
Phenolic acids (mg)	0.76 ± 0.12	0.93 ± 0.15	1.36 ± 0.27	0.074
Stilbenes (mg)	0.76 ± 0.12	0.93 ± 0.15	1.36 ± 0.27	0.074
Other polyphenols (mg)	5.67 ± 0.71	4.94 ± 0.37	5.16 ± 0.46	0.604
	**Juicy pattern**			
**Total dietary intake**				
Total energy requirement	2007.04 ± 37.99	2043.37 ± 40.02	1988.62 ± 37.97	0.595
Total energy intake (Kcal)	3027.41 ± 163.91	2731.15 ± 140.54	3137.35 ± 146.39	0.136
Total protein (%)	17.32 ± 0.53	16.63 ± 0.62	17.99 ± 0.74	0.324
Total fat (%)	29.92 ± 0.8	29.56 ± 1.00	29.65 ± 0.88	0.957
Total carbohydrates (%)	51.8 ± 1.05	51.16 ± 1.48	51.92 ± 1.2	0.899
Total fiber (g)	38.75 ± 2.46	46.71 ± 5.27	48 ± 5.37	0.290
Sugar (g)	20.94 ± 0.69	22.64 ± 1.24	23.4 ± 1.16	0.237
**Energy and nutrient intake from beverages**				
Total energy (Kcal)	467.86 ± 25.55	492.14 ± 28.03	796.81 ± 48.1	**<0.0001**
Energy (%)	18.78 ± 1.6	20.48 ± 2.21	26.5 ± 1.77	**0.010**
Protein (g)	15.62 ± 1.35	16.22 ± 1.25	18.66 ± 2.09	0.367
Carbohydrate (g)	120.4 ± 8.92	103.99 ± 7.78	111.42 ± 10.28	0.438
Fiber (g)	1.24 ± 0.15	1.18 ± 0.160	1.52 ± 0.22	0.354
Sugar (g)	115.54 ± 8.37	101.21 ± 7.42	106.78 ± 10.07	0.501
Fat (g)	13.81 ± 1.31	14.54 ± 1.21	17.03 ± 2.04	0.312
Saturated fat (g)	8.36 ± 0.82	8.86 ± 0.76	10.42 ± 1.28	0.300
Monounsaturated fat (g)	1.21 ± 0.17	1.24 ± 0.16	1.37 ± 0.23	0.812
Polyunsaturated fat (g)	0.2 ± 0.02	0.20 ± 0.02	0.23 ± 0.03	0.592
Cholesterol (mg)	44.46 ± 4.25	47.07 ± 3.92	55.22 ± 6.73	0.302
Na (mg)	313.26 ± 22.85	320.03 ± 21.77	326.09 ± 30.9	0.939
K (mg)	642.31 ± 58.88	568.90 ± 49.85	662.95 ± 74.97	0.530
Ca (mg)	501.1 ± 44.01	518.59 ± 40.10	598.27 ± 66.63	0.366
Mg (mg)	44.47 ± 4.1	38.84 ± 3.47	45.34 ± 5.05	0.503
Fe (mg)	1.07 ± 0.13	0.86 ± 0.10	1.08 ± 0.15	0.408
Zn (mg)	17.17 ± 4.59	20.43 ± 9.05	11.13 ± 2.26	0.542
P (mg)	197.5 ± 22.4	191.44 ± 21.23	216.54 ± 30.08	0.758
Se (μg)	5.38 ± 0.59	5.18 ± 0.56	5.94 ± 0.77	0.691
Vit C (mg)	78.14 ± 8.6	60.36 ± 6.09	71.47 ± 7.91	0.248
Thiamin (mg)	0.34 ± 0.03	0.36 ± 0.10	0.35 ± 0.06	0.971
Riboflavin (mg)	0.42 ± 0.04	0.42 ± 0.04	0.45 ± 0.06	0.889
Niacin (mg)	3.68 ± 0.35	3.50 ± 0.41	3.40 ± 0.27	0.855
Vit B6 (mg)	0.33 ± 0.03	0.31 ± 0.04	0.32 ± 0.04	0.958
Folate (μg)	46.42 ± 4.84	35.05 ± 3.43	41.9 ± 4.66	0.178
Choline (mg)	40.84 ± 3.73	36.8 ± 3.37	42.61 ± 4.98	0.587
Vit B12 (mg)	0.8 ± 0.09	0.83 ± 0.10	0.87 ± 0.12	0.880
Vit E (mg)	0.31 ± 0.04	0.25 ± 0.02	0.28 ± 0.03	0.358
Vit A (μg)	187.94 ± 17.77	183.64 ± 14.44	218.25 ± 24.72	0.391
Vit D (mg)	3.05 ± 0.29	3.26 ± 0.27	3.76 ± 0.40	0.286
Caffeine (mg)	44.79 ± 3.37	37.73 ± 2.87	64.47 ± 5.26	**<0.0001**
Anthocyanins (mg)	3.5 ± 0.53	3.03 ± 0.33	7.35 ± 1.27	**<0.0001**
Flavonoids (mg)	56.87 ± 5.61	62.32 ± 5.06	88.01 ± 9.17	**0.003**
Phenolic acids (mg)	1.17 ± 0.2	0.50 ± 0.08	1.38 ± 0.25	**0.003**
Stilbenes (mg)	1.17 ± 0.2	0.50 ± 0.08	1.38 ± 0.25	**0.003**
Other polyphenols (mg)	6.16 ± 0.73	4.53 ± 0.28	5.08 ± 0.48	0.089
	**Milk beverages pattern**			
**Total dietary intake**				
Total energy requirement	1987.62 ± 36.73	2013.61 ± 40.82	2038.75 ± 38.57	0.644
Total energy intake (Kcal)	2833.68 ± 155.24	2995.96 ± 133.67	3072.41 ± 161.02	0.514
Total protein (%)	17.06 ± 0.66	17.81 ± 0.61	17.01 ± 0.6	0.604
Total fat (%)	29.15 ± 0.95	30.65 ± 0.88	29.32 ± 0.78	0.421
Total carbohydrates (%)	51 ± 1.36	51.02 ± 1.21	53.15 ± 1.05	0.408
Total fiber (g)	40.4 ± 3.64	44.55 ± 4.65	49.35 ± 5.48	0.388
Sugar (g)	21.51 ± 0.86	21.56 ± 1.1	24.2 ± 1.22	0.139
**Energy and nutrient intake from beverages**				
Total energy (Kcal)	507.92 ± 31.18	459.28 ± 24.12	788.61 ± 47.1	**<0.0001**
Energy (%)	21.75 ± 2.12	17.98 ± 1.20	26.4 ± 2.11	**0.009**
Protein (g)	16.04 ± 1.18	17.57 ± 1.86	16.93 ± 1.74	0.796
Carbohydrate (g)	97.27 ± 6.83	116.61 ± 10.07	122.34 ± 9.89	0.118
Fiber (g)	1.26 ± 0.16	1.52 ± 0.21	1.17 ± 0.16	0.369
Sugar (g)	93.5 ± 6.42	111.8 ± 9.36	118.61 ± 9.87	0.103
Fat (g)	14.4 ± 1.17	16.05 ± 1.89	14.95 ± 1.59	0.752
Saturated fat (g)	8.80 ± 0.73	9.77 ± 1.18	9.09 ± 0.99	0.774
Monounsaturated fat (g)	1.18 ± 0.12	1.19 ± 0.18	1.45 ± 0.26	0.536
Polyunsaturated fat (g)	0.20 ± 0.02	0.20 ± 0.02	0.22 ± 0.03	0.764
Cholesterol (mg)	46.27 ± 3.82	52.55 ± 6.21	48.05 ± 5.17	0.673
Na (mg)	290.67 ± 19.14	346.41 ± 31.06	323.55 ± 25.07	0.296
K (mg)	571.46 ± 44.16	608.59 ± 58.00	694.73 ± 79.30	0.352
Ca (mg)	513.44 ± 38.81	561.06 ± 59.69	544.34 ± 55.00	0.803
Mg (mg)	39.66 ± 3.07	41.16 ± 3.99	47.86 ± 5.40	0.348
Fe (mg)	0.91 ± 0.11	1.08 ± 0.15	1.02 ± 0.13	0.626
Zn (mg)	15.98 ± 4.35	23.57 ± 9.40	9.36 ± 1.99	0.253
P (mg)	188.46 ± 16.63	191.77 ± 23.2	225.33 ± 32.41	0.510
Se (μg)	5.21 ± 0.48	5.25 ± 0.60	6.04 ± 0.82	0.586
Vit C (mg)	60.72 ± 6.21	71.34 ± 7.29	78.09 ± 9.08	0.263
Thiamin (mg)	0.28 ± 0.03	0.44 ± 0.11	0.34 ± 0.03	0.240
Riboflavin (mg)	0.40 ± 0.03	0.43 ± 0.04	0.46 ± 0.06	0.696
Niacin (mg)	3.15 ± 0.27	3.92 ± 0.43	3.53 ± 0.33	0.301
Vit B6 (mg)	0.30 ± 0.03	0.34 ± 0.04	0.33 ± 0.04	0.776
Folate (μg)	35.24 ± 3.5	40.43 ± 3.98	47.78 ± 5.37	0.122
Choline (mg)	37.19 ± 2.92	38.34 ± 3.78	44.74 ± 5.23	0.370
Vit B12 (mg)	0.79 ± 0.07	0.84 ± 0.11	0.86 ± 0.13	0.874
Vit E (mg)	0.24 ± 0.02	0.26 ± 0.02	0.33 ± 0.04	0.153
Vit A (μg)	185.25 ± 15.51	206.61 ± 22.62	198.34 ± 19.89	0.735
Vit D (mg)	3.29 ± 0.28	3.45 ± 0.37	3.33 ± 0.32	0.935
Caffeine (mg)	41.47 ± 3.95	45.59 ± 3.76	59.97 ± 4.36	**0.003**
Anthocyanins (mg)	3.33 ± 0.49	3.64 ± 0.39	6.91 ± 1.28	**0.003**
Flavonoids (mg)	55.62 ± 4.88	66.67 ± 6.02	85.15 ± 9.08	**0.009**
Phenolic acids (mg)	0.61 ± 0.10	0.93 ± 0.14	1.53 ± 0.28	**0.002**
Stilbenes (mg)	0.61 ± 0.10	0.93 ± 0.14	1.53 ± 0.28	**0.002**
Other polyphenols (mg)	4.75 ± 0.41	5.74 ± 0.72	5.29 ± 0.42	0.417

Vit, vitamins; Mg, magnesium; Zn, zinc; Na, sodium; K, potassium; P, phosphorus. ^a^ All nutrients were adjusted concerning energy by the residual method.

**Table 3 nutrients-15-01817-t003:** Baseline characteristics according to tertiles of beverages consumption patterns.

	Tertile 1	Tertile 2	Tertile 3	*p*-Value
**Alcoholic pattern**				
Age (years)	18.71 ± 0.10	19.13 ± 0.14	19.25 ± 0.15	1.510
BMI (kg/m^2^)	23.46 ± 0.39	22.93 ± 0.39	23.55 ± 0.43	0.496
WC (cm)	75.88 ± 1.03	74.93 ± 0.95	76.09 ± 1.01	0.678
BF (%)	27.22 ± 0.66	24.74 ± 0.75	24.57 ± 0.79	0.653
SBP (mmHg)	112.53± 1.27	112.97 ± 1.23	113.28 ± 1.15	**0.053**
DBP (mmHg)	69.39 ± 0.97	69.58 ± 0.80	69.80 ± 0.95	0.519
TG (mg/dL)	101.28 ± 7.17	89.95 ± 4.13	93.06 ± 4.05	0.062
HDL-c (mg/dL)	50.24 ± 1.17	53.30 ± 1.27	52.84 ± 1.43	**0.011**
LDL-c (mg/dL)	92.26 ± 2.38	95.84 ± 2.37	92.71 ± 1.99	0.884
GLC (mg/dL)	91.16 ± 1.64	89.47 ±0.82	88.47 ± 0.84	0.069
TC (mg/dL)	162.77 ± 3.20	167.13 ± 2.85	164.21 ± 2.38	0.575
Gender (% Female)	69	52.7	40.9	**<0.001**
Smoking status (%)	11.5	24.1	60.9	**<0.001**
Physical activity (%)	81.4	82.1	80	0.836
**Yogurt pattern**				
Age (years)	19.05 ± 0.15	19.15 ± 0.16	18.89 ± 0.10	0.390
BMI (kg/m^2^)	23.44 ± 0.4	22.94 ± 0.37	23.55 ± 0.44	0.524
WC (cm)	75.6 ± 0.94	74.51 ± 0.97	76.8 ± 1.05	0.265
BF (%)	25.75 ± 0.67	24.84 ± 0.76	25.93 ± 0.80	0.525
SBP (mmHg)	112.61 ± 1.29	112.05 ± 1.15	114.15 ± 1.11	0.481
DBP (mmHg)	68.85 ± 0.97	69.79 ± 0.85	70.12 ± 10.92	0.359
TG (mg/dL)	82.64 ± 3.98	96.94 ± 5.47	104.33 ± 6.06	**0.005**
HDL-c (mg/dL)	52.17± 1.21	53.12 ± 1.29	51.08 ± 1.40	0.630
LDL-c (mg/dL)	90.77 ± 2.27	94.61 ± 2.19	95.30 ± 2.30	0.138
GLC (mg/dL)	87.64 ± 0.76	90.39 ± 1.64	88.60 ± 0.77	0.681
TC (mg/dL)	159.51 ± 2.65	167.11 ± 2.83	167.24 ± 2.94	**0.047**
Gender (% Female)	59.1	48.7	54.9	0.493
Smoking status (%)	26.4	22.2	26.5	0.181
Physical activity (%)	84.5	77.8	81.4	0.532
**Juicy pattern**				
Age (years)	19.13 ± 0.13	18.90 ± 0.13	19.05 ± 0.15	0.759
BMI (kg/m^2^)	23.47 ± 0.41	23.26 ± 0.36	23.21 ± 0.44	0.891
WC (cm)	75.89 ± 0.99	75.83 ± 0.93	75.19 ± 1.05	0.860
BF%	25.58 ± 0.83	25.81 ± 0.66	25.10 ± 0.75	0.644
SBP (mmHg)	114.65 ± 1.23	111.93 ± 1.17	112.21 ± 1.15	0.115
DBP (mmHg)	70.19 ± 0.87	68.10 ± 0.91	70.50 ± 0.95	0.791
TG (mg/dL)	93.24 ± 4.40	96.07 ± 4.92	95.00 ± 6.47	0.826
HDL-c (mg/dL)	52.14 ± 1.29	52.04 ± 1.32	52.22 ± 1.32	0.981
LDL-c (mg/dL)	96.33 ± 2.14	92.35 ± 2.13	92.12 ± 2.47	0.195
GLC (mg/dL)	88.26 ± 0.72	87.58 ± 0.70	90.88 ± 1.73	0.103
TC (mg/dL)	167.34 ± 2.56	163.42 ± 2.80	163.34 ± 3.10	0.320
Gender (%Female)	50.4	61.4	50.4	0.945
Smoking status (%)	27.4	22.8	24.8	0.707
Physical activity (%)	79.6	80.7	83.2	0.518
**Milk beverages pattern**				
Age (years)	19.18 ± 0.14	18.86 ± 0.12	19.05 ± 0.15	0.499
BMI (kg/m^2^)	22.7 ± 0.39	23.43 ± 0.39	23.83 ± 0.43	0.132
WC (cm)	74.43 ± 0.97	75.73 ± 0.93	76.77 ± 1.07	0.244
BF%	26.11 ± 0.77	25.74 ± 0.71	24.64 ± 0.75	0.395
SBP (mmHg)	112.20 ± 1.06	112.83 ± 1.29	113.75 ± 1.19	0.632
DBP (mmHg)	70.71 ± 0.81	68.52 ± 0.96	69.56 ± 0.95	0.258
TG (mg/dL)	90.36 ± 5.40	97.59 ± 4.85	96.34 ± 5.70	0.572
HDL-c (mg/dL)	52.49 ± 1.34	50.94 ± 1.13	52.98 ± 1.44	0.633
LDL-c (mg/dL)	94.38 ± 2.24	93.66 ± 2.50	92.75 ± 2.01	0.656
GLC (mg/dL)	88.19 ± 0.78	88.90 ± 0.88	89.62 ± 1.62	0.558
TC (mg/dL)	164.94 ± 2.88	164.12 ± 3.06	165.03 ± 2.54	0.939
Gender (%Female)	56.6	57	48.7	0.200
Smoking status (%)	26.5	23.7	24.8	0.801
Physical activity (%)	84.1	85.1	74.3	0.063

BF%: body fat percent; SBP: systolic blood pressure; DBP: diastolic blood pressure; TG: triglycerides; HDL-c: high-density lipoprotein cholesterol; LDL-c: low-density lipoprotein cholesterol; GLC: glucose; TC: total cholesterol.

**Table 4 nutrients-15-01817-t004:** Unadjusted and adjusted odds ratio (95% confidence intervals) for cardiovascular risk factors across tertiles of beverage patterns.

	Model ^a^	Tertile 1	Tertile 2	Tertile 3
**Alcoholic pattern**				
High WC	1	1(REF)	0.695 (0.354–1.367)	0.948 (0.498–1.802)
	2	1(REF)	0.944 (0.311–2.863)	0.626 (0.216–1.813)
	3	1(REF)	1.166 (0.138–9.828)	2.815 (0.394–20.108)
High %BF	1	1(REF)	**0.570 (0.335–0.971)**	0.724 (0.428–1.225)
	2	1(REF)	0.597 (0.298–1.199)	**0.371 (0.173–0.798)**
	3	1(REF)	0.561 (0.225–1.400)	0.544 (0.209–1.416)
High BP	1	1(REF)	1.145 (0.541–2.421)	0.498 (0.202–1.226)
	2	1(REF)	1.159 (0.523–2.565)	**0.318 (0.116–0.871)**
	3	1(REF)	0.742 (0.262–2.105)	0.327 (0.092–1.160)
High TG	1	1(REF)	0.628 (0.269–1.464)	0.498 (0.202–1.226)
	2	1(REF)	0.693 (0.280–1.715)	0.402 (0.154–1.049)
	3	1(REF)	0.277 (0.060–1.277)	0.243 (0.048–1.238)
Low HDL	1	1(REF)	0.814 (0.472–1.402)	0.701 (0.403–1.218)
	2	1(REF)	0.868 (0.494–1.524)	0.679 (0.379–1.217)
	3	1(REF)	0.772 (0.367–1.622)	0.557 (0.246–1.259)
High LDL	1	1(REF)	1.548 (0.902–2.658)	1.126 (0.649–1.952)
	2	1(REF)	1.522 (0.880–2.631)	1.067 (0.603–1.889)
	3	1(REF)	1.516 (0.751–3.061)	0.974 (0.434–2.187)
High GLC	1	1(REF)	0.809 (0.335–1.956)	0.556 (0.210–1.468)
	2	1(REF)	0.820 (0.329–2.048)	0.480 (0.173–1.333)
	3	1(REF)	**0.232 (0.061–0.875)**	0.446 (0.101–1.975)
High COL	1	1(REF)	0.990 (0.425–2.308)	0.908 (0.383–2.152)
	2	1(REF)	1.075 (0.457–2.527)	0.963 (0.400–2.318)
	3	1(REF)	0.668 (0.225–1.980)	0.610 (0.173–2.155)
**Yogurt pattern**				
High WC	1	1(REF)	0.650 (0.328–1.289)	1.000 (0.529–1.892)
	2	1(REF)	0.639 (0.217–1.884)	0.997 (0.323–3.082)
	3	1(REF)	0.374 (0.045–3.089)	0.444 (0.062–3.176)
High %BF	1	1(REF)	0.794 (0.469–1.343)	0.721 (0.424–1.224)
	2	1(REF)	0.801 (0.402–1.598)	0.518 (0.248–1.081)
	3	1(REF)	0.903 (0.371–2.195)	0.492 (0.188–1.288)
High BP	1	1(REF)	1.682 (0.728–3.884)	1.457 (0.618–3.432)
	2	1(REF)	1.533 (0.630–3.730)	1.266 (0.521–3.073)
	3	1(REF)	0.940 (0.296–2.984)	0.839 (0.271–2.595)
High TG	1	1(REF)	0.990 (0.396–2.480)	1.457 (0.618–3.432)
	2	1(REF)	1.135 (0.431–2.991)	1.224 (0.490–3.056)
	3	1(REF)	0.702 (0.144–3.427)	1.020 (0.274–3.794)
Low HDL	1	1(REF)	1.112 (0.639–1.934)	1.172 (0.674–2.036)
	2	1(REF)	1.157 (0.655–2.043)	1.297 (0.723–2.327)
	3	1(REF)	1.182 (0.545–2.561)	0.957 (0.441–2.077)
High LDL	1	1(REF)	0.815 (0.475–1.401)	0.963 (0.564–1.646)
	2	1(REF)	0.786 (0.453–1.366)	0.897 (0.518–1.553)
	3	1(REF)	0.768 (0.369–1.600)	0.940 (0.446–1.980)
High GLC	1	1(REF)	0.386 (0.131–1.134)	1.000 (0.429–2.331)
	2	1(REF)	**0.317 (0.102–0.987)**	0.805 (0.329–1.974)
	3	1(REF)	**0.110 (0.022–0.559)**	0.575 (0.169–1.957)
High COL	1	1(REF)	0.581 (0.231–1.460)	1.000 (0.429–2.331)
	2	1(REF)	0.634 (0.247–1.626)	1.117 (0.494–2.526)
	3	1(REF)	0.563 (0.157–2.014)	1.112 (0.380–3.255)
**Juicy pattern**				
High WC	1	1(REF)	1.305 (0.524–3.253)	1.344 (0.577–3.133)
	2	1(REF)	1.408 (0.320–6.196)	2.224 (0.397–12.442)
	3	1(REF)	2.300 (0.428–12.356)	1.769 (0.254–12.313)
High %BF	1	1(REF)	1.015 (0.520–1.982)	0.854 (0.431–1.692)
	2	1(REF)	0.934 (0.369–2.362)	0.444 (0.161–1.225)
	3	1(REF)	1.006 (0.373–2.717)	0.413 (0.145–1.173)
High BP	1	1(REF)	1.200 (0.435–3.313)	0.889 (0.321–2.459)
	2	1(REF)	1.042 (0.350–3.100)	0.854 (0.294–2.482)
	3	1(REF)	1.069 (0.350–3.260)	0.818 (0.275–2.430)
High TG	1	1(REF)	0.209 (0.043–1.005)	3.242 (0.631–16.665)
	2	1(REF)	**1.155 (1.027–2.885)**	4.444 (0.715–27.629)
	3	1(REF)	**1.084 (1.011–2.656)**	5.974 (0.798–44.73)
Low HDL	1	1(REF)	1.380 (0.681–2.797)	0.682 (0.331–1.405)
	2	1(REF)	1.284 (0.620–2.660)	0.702 (0.329–1.495)
	3	1(REF)	1.386 (0.653–2.941)	0.731 (0.333–1.602)
High LDL	1	1(REF)	1.136 (0.568–2.274)	0.842 (0.415–1.706)
	2	1(REF)	1.200 (0.579–2.486)	0.750 (0.354–1.586)
	3	1(REF)	1.201 (0.574–2.512)	0.697 (0.324–1.502)
High GLC	1	1(REF)	1.242 (0.396–3.900)	1.181 (0.403–3.459)
	2	1(REF)	1.105 (0.312–3.910)	1.181 (0.374–3.727)
	3	1(REF)	4.083 (0.841–19.817)	1.088 (0.321–3.694)
High COL	1	1(REF)	0.667 (0.224–1.984)	1.205 (0.383–3.790)
	2	1(REF)	0.690 (0.225–2.119)	1.210 (0.368–3.976)
	3	1(REF)	0.716 (0.228–2.250)	1.358 (0.382–4.829)
**Milk beverages pattern**				
High WC	1	1(REF)	0.821 (0.337–1.997)	1.895 (0.790–4.548)
	2	1(REF)	1.240 (0.284–5.404)	1.309 (0.229–7.469)
	3	1(REF)	0.927 (0.178–4.819)	4.725 (0.351–63.667)
High %BF	1	1(REF)	0.662 (0.344–1.274)	1.224 (0.611–2.455)
	2	1(REF)	0.642 (0.270–1.526)	0.799 (0.288–2.217)
	3	1(REF)	0.668 (0.273–1.638)	0.848 (0.294–2.442)
High BP	1	1(REF)	0.634 (0.239–1.677)	0.991 (0.324–3.027)
	2	1(REF)	0.612 (0.210–1.788)	0.844 (0.263–2.705)
	3	1(REF)	0.608 (0.206–1.798)	0.674 (0.199–2.281)
High TG	1	1(REF)	0.918 (0.254–3.312)	1.965 (0.591–6.535)
	2	1(REF)	0.899 (0.244–3.315)	1.759 (0.477–6.488)
	3	1(REF)	0.645 (0.152–2.731)	1.871 (0.470–7.457)
Low HDL	1	1(REF)	0.837 (0.412–1.700)	1.614 (0.780–3.339)
	2	1(REF)	0.803 (0.384–1.682)	1.687 (0.782–3.638)
	3	1(REF)	0.766 (0.359–1.633)	1.858 (0.835–4.134)
High LDL	1	1(REF)	0.698 (0.356–1.372)	1.069 (0.516–2.212)
	2	1(REF)	0.717 (0.360–1.43)	1.030 (0.483–2.196)
	3	1(REF)	0.673 (0.331–1.369)	1.083 (0.500–2.344)
High GLC	1	1(REF)	1.061 (0.364–3.091)	0.991 (0.324–3.027)
	2	1(REF)	1.083 (0.351–3.338)	0.883 (0.260–3.005)
	3	1(REF)	**5.304 (1.292–21.773)**	0.932 (0.251–3.467)
High COL	1	1(REF)	0.789 (0.271–2.298)	1.365 (0.451–4.131)
	2	1(REF)	0.797 (0.270–2.349)	1.510 (0.483–4.716)
	3	1(REF)	0.809 (0.268–2.445)	1.835 (0.541–6.219)

WC: waist circumference; BF%: body fat percent; SBP: systolic blood pressure; DBP: diastolic blood pressure; TG: triglycerides; HDL-c: high-density lipoprotein cholesterol; LDL-c: low-density lipoprotein cholesterol; GLC: glucose; TC: total cholesterol; REF: reference value. Bolding denotes statistically significant results. ^a^ Model 1 was unadjusted. Model 2 was adjusted for gender, smoking status, and BMI. Model 3 was adjusted for all covariates in model 2 plus total energy intake, and percentage of energy from protein, fat, and carbohydrates.

## Data Availability

Data available on request due to restrictions in privacy or ethical issues. The data presented in this study are available on request from the corresponding author. The data are not publicly available due to confidentiality and ethical issues.

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
