# Peer review of "Beverage Consumption Patterns and Nutrient Intake Are Associated with Cardiovascular Risk Factors among Urban Mexican Young Adults"

_nutrients, 2023, doi:10.3390/nu15081817_

Round 1

Reviewer 1 Report

The authors investigate the role of beverage consumption and nutrition intake in urban Mexican young adults and its correlation with CV risk. The Introduction is short. Methods are not adequate. General diet of the subjects has not been described. neither in total calories or in macronutrients intake. This represents a major defect that must be addressed. The definition of CV risk factors miss the anthropometric features of the patients. Patient final sample is small for correlation analyses. Moreover, in the studied patterns, as already mentioned, diet is missing as well as physical activity and calorie expenditure of the patients.

Overall the paper requires extensive revisiting as there are major flaws and the need for additional data and analyses.

Author Response

The authors investigate the role of beverage consumption and nutrition intake in urban Mexican young adults and its correlation with CV risk. The Introduction is short. Methods are not adequate. General diet of the subjects has not been described. neither in total calories or in macronutrients intake. This represents a major defect that must be addressed. The definition of CV risk factors miss the anthropometric features of the patients. Patient final sample is small for correlation analyses. Moreover, in the studied patterns, as already mentioned, diet is missing as well as physical activity and calorie expenditure of the patients.

Overall the paper requires extensive revisiting as there are major flaws and the need for additional data and analyses.

Response 1: It was considered all your observations:

The Introduction was improved.

Methods were corrected.

General diet of the subjects, both in total energy and macronutrientes intake were included and takinng into multivariate logistic regression models, possible immportant cofounder.

Anthropometric features of the patients, such as BMI and waits circumfernces were included into the definition of CV.

Correlation analysis was eliminated from the analysis. Instead, ANOVA were conducted to test for significant differences in both mean of total energy and nutrient intake, and mean values of anthropometric, biochemical and nutritional variables.

Physical activity and energy expenditure of the participants were included.

Reviewer 2 Report

The manuscript submitted by Salinas-Mandujano et al., titled: "Beverage consumption patterns and nutrient intake are associated with cardiovascular risk factors among urban Mexican young adults" is an interesting work investigating the association of beverage consumption/nutrient intake and risk for CVD in young Mexican adults.

the reviewer would like to bring the following points to the authors' attention:

1. What were the inclusion and exclusion criteria for participation in the study? Please specify.

2. How was the number of participants determined? (eg power calculation or previous knowledge/experience).

3. What was the hypothesis for the work presented?

4. An interesting point to discuss in the context on beverage preference is the various non-alcoholic types of beverages that are not dairy based and may have potential for health benefits. These could offer a potential healthier alternative to the population studied and thus an interesting point to explore. An interesting review to consider for a short discussion is the following: 

Sikalidis, A.K.; Kelleher, A.H.; Maykish, A.; Kristo, A.S. Non-Alcoholic Beverages, Old and Novel, and Their Potential Effects on Human Health, with a Focus on Hydration and Cardiometabolic Health. Medicina 2020, 56, 490. https://doi.org/10.3390/medicina56100490.

Author Response

Response to Reviewer 2 Comments

Point 1: What were the inclusion and exclusion criteria for participation in the study? Please specify.

Response 1: The inclusion criteria used were: students between 18 and 24 years of age, in their first year of school. The exclusion criteria were: participants with current clinical evidence of infectious disease, anemia, pregnancy or lactation, current medical or nutritional treatment, as well as those with missing information derived from dietary or beverage questionnaire, as well as  biochemical and anthropometric  measurements. This information was included in Material and methods section (Pg. 3)

Point 2:. How was the number of participants determined? (eg power calculation or previous knowledge/experience).

Response 2:  Subjects (n=1,160) in the study were recruited by a non probabilistic sample, including all the freshmen attending to the Autonomous University of Queretaro, who voluntarily agreed to participate in the study

Point 3:. What was the hypothesis for the work presented?

Response 3: In our study, the hypothesis tested was that beverages consumption patterns with high consumption of SSBs would be associated to greater energy intake, obesity, and CVRFs, compared to those with low consumption of SSBs. This information was included in Introduction section (Pg. 2).

Point 4: An interesting point to discuss in the context on beverage preference is the various non-alcoholic types of beverages that are not dairy based and may have potential for health benefits. These could offer a potential healthier alternative to the population studied and thus an interesting point to explore. An interesting review to consider for a short discussion is the following:

Sikalidis, A.K.; Kelleher, A.H.; Maykish, A.; Kristo, A.S. Non-Alcoholic Beverages, Old and Novel, and Their Potential Effects on Human Health, with a Focus on Hydration and Cardiometabolic Health. Medicina 2020, 56, 490. https://doi.org/10.3390/medicina56100490.

Response 4: This information was included in Discussion section (Pg. 17).

Round 2

Reviewer 1 Report

The authors have extensively modified the article. The required adjustments have been made. The article is now fit.

Reviewer 2 Report

The authors have made a reasonable effort in addressing reviewer's comments.